# *Burkholderia Phytofirmans* PsJN Stimulate Growth and Yield of Quinoa under Salinity Stress

**DOI:** 10.3390/plants9060672

**Published:** 2020-05-26

**Authors:** Aizheng Yang, Saqib Saleem Akhtar, Qiang Fu, Muhammad Naveed, Shahid Iqbal, Thomas Roitsch, Sven-Erik Jacobsen

**Affiliations:** 1School of Water Conservancy and Civil Engineering, Northeast Agriculture University, Harbin 150030, China; aizheng.yang@neau.edu.cn; 2Department of Plant and Environmental Sciences, Faculty of Science, University of Copenhagen, Højbakkegård Allé 13, DK-2630 Tåstrup, Denmark; sasa@plen.ku.dk (S.S.A.); roitsch@plen.ku.dk (T.R.); 3Dansk Agro Aps, Snubbekorsvej 20 C, DK-2630 Tåstrup, Denmark; 4Institute of Soil and Environmental Sciences, University of Agriculture Faisalabad, Faisalabad 38040, Pakistan; muhammad.naveed@uaf.edu.pk; 5Department of Agronomy, Muhammad Nawaz Shareef University of Agriculture, Multan 66000, Pakistan; shahid.iqbal@mnsuam.edu.pk; 6Department of Adaptive Biotechnologies, Global Change Research Institute, CAS, 603 00 Brno, Czech Republic; 7Quinoa Quality ApS, DK-4420 Regstrup, Denmark

**Keywords:** endophytic bacteria, plant growth promoting bacteria (PGPB)

## Abstract

One of the major challenges in agriculture is to ensure sufficient and healthy food availability for the increasing world population in near future. This requires maintaining sustainable cultivation of crop plants under varying environmental stresses. Among these stresses, salinity is the second most abundant threat worldwide after drought. One of the promising strategies to mitigate salinity stress is to cultivate halotolerant crops such as quinoa. Under high salinity, performance can be improved by plant growth promoting bacteria (PGPB). Among PGPB, endophytic bacteria are considered better in stimulating plant growth compared to rhizosphere bacteria because of their ability to colonize both in plant rhizosphere and plant interior. Therefore, in the current study, a pot experiment was conducted in a controlled greenhouse to investigate the effects of endophytic bacteria i.e., *Burkholderia phytofirmans* PsJN on improving growth, physiology and yield of quinoa under salinity stress. At six leaves stage, plants were irrigated with saline water having either 0 (control) or 400 mM NaCl. The results indicated that plants inoculated with PsJN mitigated the negative effects of salinity on quinoa resulting in increased shoot biomass, grain weight and grain yield by 12%, 18% and 41% respectively, over un-inoculated control. Moreover, inoculation with PsJN improved osmotic adjustment and ion homeostasis ability. In addition, leaves were also characterized for five key reactive oxygen species (ROS) scavenging enzyme in response to PsJN treatment. This showed higher activity of catalase (CAT) and dehydroascobate reductase (DHAR) in PsJN-treated plants. These findings suggest that inoculation of quinoa seeds with *Burkholderia phytofirmans* PsJN could be used for stimulating growth and yield of quinoa in highly salt-affected soils.

## 1. Introduction

Changing climatic conditions and misuse of agricultural land over the last few decades has led to an increase in the area of salt-affected soils [1]. Worldwide, the Food and Agriculture Organization (FAO) [1] estimates that around 1.2 billion hectares of land are affected by salinity. High salinity affects plant growth and performance, challenging agricultural production either by enhancing osmotic potential or by specific ion toxicity [2]. Different strategies have been used to counter salinity stress and improve the performance of plants. These are either related to soil management such as compost application, green manuring and soil amendment such as biochar, or plant management such as the cultivation of halophytes, use of microbial technology and developing transgenic plants [3,4,5,6,7,8,9]. It is regarded as a long-term process to rehabilitate saline soils by the use of a soil management strategy. Limited success has also been achieved by the use of breeding with transgenic technology, due to the genetic complexity of salinity and other abiotic stress tolerance. 

Halophytic crops like quinoa (*Chenopodium quinoa* Willd.) have special features to survive under salinity stress and adapt to saline environment during the entire growth stage [10,11]. Quinoa has proved to be one of the most salt-tolerant crops in existence, and may be the most [12,13,14,15], due to a range of important mechanisms, such as the effect of the leaf bladders and other internal strategies [16,17]. Not only their own adaptive mechanism, but also rhizobacteria play a vital role in alleviating salinity stress symptoms. Recently, studies have shown that bacteria especially from the rhizosphere may interact with plants and affect growth. Plant growth-promoting bacteria (PGPB) can mitigate negative effect induced by abiotic stress on plants and enhance adaptation to a harsh growth environment by increasing metabolic activity [18]. Some PGPB can penetrate into roots and even move to the stem. Moreover, PGPB have direct and indirect mechanisms to stimulate plant growth and alleviate salinity stress. Furthermore, PGPB can directly promote atmospheric nitrogen fixation, phosphate solubilization and siderophore production [19,20]. In particular, PGPB can modulate plant phytohormone level such as gibberellins (GA) and abscisic acid (ABA) in plant tissue, which can alleviate salt stress in crops [21,22]. Also, certain PGPB is associated with the production of auxins (indole-3-acetic acid, IAA) and the enzyme 1-aminocyclopropane-1-carboxylate (ACC)-deaminase, which can cleave ethylene and are crucial signaling molecules for maintaining plant growth and triggering defense mechanism [23]. For indirect aspects, PGPB could enhance the ability to defense against pathogens and insect herbivore for plants by adhesion to the whole plant body [24].

Endophytic bacteria may play an important role for plants to survive and adapt to harsh environments, because they are intimately associated with plant tissues, affecting plant growth. However, very few studies in this regard in halophytes have been reported. In addition, very little is known about the antioxidant activity of halophytes inoculated with endophytic bacteria. Therefore, the present study was conducted to understand the mechanism of endophytic bacterium. *Burkholderia phytofirmans* PsJN for inducing salt tolerance in quinoa by stimulating growth, physiology and yield under salinity stress. Our results would contribute to further improve salt tolerance and the productivity of halophytes and saline soil utilization.

## 2. Results

### 2.1. Growth and Yield Responses

Table 1 shows all growth and yield parameters were significantly decreased by salinity stress (*p* < 0.001). In addition, plant height and panicle length were remarkably affected by interaction between salinity and inoculation (*p* < 0.05) and 100-grain weight was solely affected by salinity. Moreover, shoot biomass and grain yield were significantly affected by inoculation (*p* < 0.05). Quinoa inoculated with PsJN had 11.9% and 41.4% increase in shoot biomass and grain yield respectively than un-inoculated treatment under 400 mM NaCl.

### 2.2. Leaf Stomatal Conductance (gs) and Photosynthetic Rate (An)

Data of leaf photosynthetic rate (A_n_) and stomatal conductance (g_s_) are presented in Figure 1. Both A_n_ and g_s_ decreased significantly under salinity stress and were significantly affected by both salinity and inoculation (*p* < 0.05). Inoculation of plants with PsJN improved 21.6% and 36.0% the A_n_ and g_s_ under saline irrigation compared to the un-inoculated control, respectively. Furthermore, no significant effect of PsJN was noticed in A_n_ and g_s_ under non-saline irrigation.

### 2.3. Plant–Water Relations

Figure 2 indicates the response of PsJN on plant–water relations under saline and non-saline irrigations. Relative water content (RWC), leaf water potential (Ψ_leaf_), osmotic potential (Ψ_π_) and turgor potential (Ψ_p_) were significantly reduced in saline irrigation compared to non-saline irrigation. However, inoculation of plant with PsJN improved Ψ_leaf_ (less negative), Ψ_π_ (less negative), Ψ_p_ and RWC compared to control under salinity stress. Furthermore, no effect of PsJN was noticed on all parameters under non-saline irrigation. 

### 2.4. Leaf Na^+^ and K^+^

Figure 3 indicates the response of PsJN on leaf Na^+^ and K^+^ content under saline and non-saline irrigations. Both leaf Na^+^ and K^+^ contents were increased under saline than non-saline irrigation, respectively. However, under saline irrigation, significant reduction in leaf Na^+^ and improved leaf K^+^ content was observed in PsJN inoculated plant compared to un-inoculated control. PsJN inoculation induced leaf Na^+^ content decreasing 37.4% and K^+^ content increasing 22.6%, respectively, compared to the control under saline irrigation.

### 2.5. Leaf Abscisic Acid (ABA) Concentration

Saline irrigation had a significant effect on increasing leaf ABA concentrations (Figure 4). Under saline irrigation, PsJN inoculation significantly decreased the ABA concentration compared to the un-inoculated control while no difference was observed under non-saline irrigations.

### 2.6. Key Reactive Oxygen Species (ROS) Scavenging Enzyme Response

In the current study, five central reactive oxygen species (ROS) scavenging enzymes were studied from quinoa leave samples. The data is presented in the form of heat map with different color intensities ranging from dark red to dark blue (Table 2). Maximum enzyme activity was presented with dark red while lowest with dark blue. The data indicated that, out of the five studied ROS scavenging enzymes, only catalase (CAT) and dehydroascobate reductase (DHAR) responded positively (higher activity) to PsJN treatment compared to un-inoculated control. While CAT activity was significantly higher in PSJN treatment under salt stress compared to un-inoculated control.

### 2.7. Colony-Forming Units (CFU) of Burkholderia Phytofirmans PsJN in Rhizosphere and Root Interior

The inoculum strain PsJN efficiently colonized rhizosphere, root and shoot interior of quinoa under 0 and 400 mM NaCl solution (Figure 5). A viable count (colony-forming units (CFU) g^−1^ dry mass) of PsJN in root interior of quinoa under non-saline irrigation was 2.3 times higher than under saline irrigation.

### 2.8. Salt Tolerance of PsJN

The cell growth of bacterial strain PsJN under different saline irrigation was observed by spectrophotometer (Figure 6). The cell growth of PsJN remained almost stable until 100 mM NaCl. Thereafter, the growth of PsJN sharply decreased at 100–200 mM NaCl, then decreased gradually with increasing NaCl concentrations. A dramatic decline in cell growth of PsJN was noticed at 800–1000 NaCl.

### 2.9. Principal Component Analysis

The PC 1 and PC 2 of principal component analysis (PCA) represented 83.7% of the total variables in data profile (Figure 7). The clustered same scatter points but not intersect with others indicated there were significant differences among treatments. The PC1 (74.4% of data variability) separated treatments according to salinity stress, one group consisted of 0 mM NaCl treatments and the other group of 400 mM NaCl treatments. The variables involved in mediating physiological responses to salinity stress showed positively correlation with PC1 such as ABA, leaf potential and osmotic potential. The PC2 (9.3% of data variability) divided treatments depending on PsJN inoculation. In addition, the angle between two variables indicates correlation between them, acute and obtuse angles indicate positive and negative correlation, respectively.

## 3. Discussion

Nowadays, soil salinity is eroding 3 hectares of cultivated land every minute and, in addition, improper drainage and irrigation management have further exacerbated the secondary salinization of the soil [25]. Cultivation of halophytes such as quinoa under salt-affected conditions could be a wise choice to improve global food security. However, the growth and performance of quinoa is also affected under extreme soil salinization. On the other hand, certain PGPB have the ability to survive and promote plant growth under extreme soil salinization [18]. Among PGPB, endophytic bacteria offer advantages over rhizosphere bacteria as these have the capability to modify plant biochemistry by residing in plant tissue [26]. Therefore, here in the current study we evaluated the effect of growth promoting endophytic bacteria such as *Burkholderia phytofirmans* PsJN on improving growth, physiology, antioxidant activity and yield of quinoa under salinity stress.

Salinity stress as a vital abiotic stress leads to stomatal closure-induced CO_2_ deficiency resulting in reduced CO_2_ fixation [27]. This negative effect eventually causes excess production of various ROS. ROS play important roles in cell signaling and homeostasis while over-production of ROS could significantly disrupt cell functions and even damage cell structures by membrane lipid peroxidation and protein degradation [28]. Hence, to avoid destructive oxidative reactions by excessive ROS accumulation, plants activate antioxidant defense systems including non-enzymatic and enzymatic antioxidants such as superoxide dismutase (SOD), catalase (CAT), peroxidase (POX), DHAR and glutathione reductase (GR) [29]. SOD catalyze the dismutation of superoxide (O^2−^) into hydrogen peroxide (H_2_O_2_), thus providing protection against oxidative stress in plants [30]. Abundant amounts of CAT and POX are involved in H_2_O_2_ scavenging while decomposing it into water and oxygen. DHAR and GR perform antioxidant functions in the ascorbate–glutathione cycle [31,32]. In addition, soil salinity inhibits the ability of plants to absorb water and nutrients, and also may cause excess Na^+^ and Cl^−^ accumulations that can reach toxic levels [33]. Moreover, salt stress induces osmotic stress triggering a reduced water potential with increased ABA concentration, therefore photosynthesis would be drastically reduced [34].

Halophytes have several mechanisms to handle salinity stress, however, soil salinity still affect their metabolism and physiological processes in a certain way [35,36]. High ion concentrations such as Na^+^ and Cl^−^ in plant tissue not only cause osmotic stress but also toxic effects. Few halophyte species grow without significant plant yield under salinity stress which correlates negatively with increasing internal Na^+^ concentrations. This is also corroborated by PCA analysis in the present study, we detected a negative correlation between Na^+^ content with growth and yield parameters (Table 1 and Figure 7). Quinoa exhibits a marked preference for Na^+^ over K^+^ and has quantitatively higher Na^+^ absorption (Figure 3). In our case, leaf Na^+^ content increased dramatically under salinity stress, but still maintain a higher K^+^/ Na^+^ ratio in leaves. Flowers and Colmer [37] and Shabala and Pottosin [38] reported that halophytes relied on inorganic ion sequestration (e.g., Na^+^ and Cl^−^) in the vacuoles of cells and K^+^ retention to maintain cell turgor potential and thus moderate K^+^/Na^+^ ratio under salinity. Apart from this, we detected that PsJN inoculated plant reduced Na^+^ influx and enhanced K^+^ uptake. This phenomenon could be achieved by a sodium hydrogen exchanger 2 (NHX2), a vacuolar antiporter that expel excess Na^+^ from cells or compartmentalize into the vacuole and thus enhance K^+^/Na^+^ ratio [39]. 

In order to maintain water flow from the soil to the roots, plants need to adjust osmotic potential to absorb water under salinity stress [40]. We noticed that lower water potential (Ψ_leaf_) is coupled with decreased osmotic potential (Ψ_π_) to maintain moderate relative water content (RWC) and turgor pressure (Ψ_p_) in plants (Figure 2). Apart from this, we detected that PsJN inoculation could regulate the plant cell water relations effectively to alleviate negative effects under salinity stress resulting in relatively higher photosynthetic rate (A_n_) and yield (Figure 1 and Table 1). This is due to the ability of PsJN to accumulate proline faster to regulate Ψ_π_ [41]. On the other hand, PsJN strain inoculated plants showed an accelerated up-regulation for abiotic stress responsive genes and triggered mechanisms of plant metabolism earlier [39]. As mentioned previously, PsJN strain induced a greater decrease in leaf Na^+^ content which was related to an increase in Ψ_π_ thus mitigating osmotic stress. In the PCA analysis, leaf Na^+^ content clustered together with Ψ_π_ indicating correlation between them (Figure 7).

An understanding of the effect of the inoculation of a halophyte, quinoa, with PGPB, on salinity stress, is important to further increase the adaptation of quinoa to saline areas. ABA is a phytohormone that plays a fundamental role in mitigating plant responses to salinity stress including changes in stomatal conductance and many stress-associated gene expressions [42,43]. High solute concentration triggers ABA accumulation as a result of physiological dehydration and osmotic stress, accompanying extremely negative water relationship under salinity stress [44]. Consistent with this, we detected a positive correlation between ABA and leaf Na^+^, and in contrast a negative correlation between ABA and plant water relation parameters (RWC and Ψ_p_), also confirmed by PCA analysis (Figure 7). Moreover, ABA mediates stomata closure that inhibit gas exchange and carbon dioxide assimilation resulting in a reduced A_n_ and yield under salinity stress [45]. In the present study, this is proved by PCA analysis where obtuse angles were shown between ABA with g_s_, A_n_, yield and composition parameter respectively, indicating negative correlation with them. However, a decreased ABA level was noted in PsJN-inoculated plants under salinity stress. On the one hand, this is due to the ability of PsJN to control Na^+^ homeostasis by improving Na^+^ exclusion thus mitigating osmotic stress in plants. On the other hand, the PsJN strain might reduce the ABA level by affecting ABA biosynthetic gene expression [46]. 

Halophytes possess a complex and efficient antioxidant defense system (both non-enzymatic and enzymatic antioxidants) which can scavenge excessive ROS and thus avoid cellular oxidative damage [35]. Moreover, Srivastava et al. [47] concluded that halophyte enzymes were comparably more resistant and stable to maintain redox homeostasis under stress. However, in this regard there is still much debate, as Kumari et al. [48] found halophytes may not require a high level of antioxidant activity due to their ability to limit ROS over-accumulation by efficient mechanisms of Na^+^ exclusion. We have used an semi-high throughput analytical platform that allows the determination of up to nine different antioxidative enzymes from the very same extract at a microtiter plate scale [49]. In the present study, we only detected significant increase in CAT activity in PsJN inoculated treatment under salinity stress, whereas there were no significant fluctuations in SOD, DHAR, GR and POX activity (Table 2). This result implies the high turnover rate of CAT which can catalyze the decomposition of H_2_O_2_ into water and oxygen more efficiently and effectively, thus limiting the induction of other antioxidants. Similarly, Jithesh et al. [50] reported that one unit of CAT protein complex can decompound millions of H_2_O_2_ molecules per second. In addition, PsJN strain accelerates the accumulation of antioxidant enzymes [51,52]. In contrast, the growth promotion of maize by *Bacillus licheniformis* FMCH 001 was found to have only a very limited impact on antioxidant metabolism [53].

These results of the determination of an antioxidant enzyme activity signature show also the relevance to implement a cell physiological phenotyping into a holistic phenomics approach [54,55] and future work and analysis of these findings will help to give more clarity on the underlying mechanistic effect of the bacterial inoculation on regulating plant physiology in a systemic way.

## 4. Materials and Methods 

### 4.1. Collection of Bacterial Strains 

Bacterial strain *Burkholderia phytofirmans* PsJN was kindly provided by the AIT Austrian Institute of Technology GmbH, Austria. This strain has been successfully reported to enhance the growth of glycophytic plants (maize, wheat) under normal and stress conditions [56,57]. However, the effect of PsJN on halophytic crops such as quinoa has not been tested before.

### 4.2. Inoculum Preparation and Seed Bacterization

Inocula of *Burkholderia phytofirmans* PsJN was prepared as described in our previous study [57]. In brief, strains PsJN::gusA 10 was cultured in 250 mL Erlenmeyer flasks containing Luria–Bertani (LB) broth containing spectinomycin (100 mg mL^−1^) at 28 °C for 48 h in an orbital shaking incubator (SunGene GmbH, Innova 4430, NJ, USA) at 180 rev min^−1^. For the un-inoculated control, sterilized broth was incubated under the same conditions. Prior to seed inoculation, optical density of bacterial culture was measured by using spectrophotometer (GENESYS 10, Thermo Spectronic, NY, USA) and adjusted at 0.5 to obtain a uniform bacterial population [10^8^–10^9^ CFU mL^−1^] for inoculation.

For seed inoculation, quinoa seed were first surface sterilized by dipping in 70% ethanol for 60 s and then treated with 3% sodium hypochlorite for 5 min followed by five washing with sterilized milli-Q water. Thereafter, surface-disinfected seeds were dipped in 20 mL of the bacterial suspension for 2 h. 

### 4.3. Pot Study

A pot experiment was conducted in greenhouse at University of Copenhagen, Denmark, to evaluate the effectiveness of *Burkholderia phytofirmans* PsJN on improving growth, physiology and yield of quinoa cv. Titicaca under salt stress.

Five inoculated seeds were sown in each plastic pot (height 10 cm, diameter 4 cm) containing 5 kg of mixture of peat and sandy loam soil (3:2). After one week of germination, seedlings were removed to obtain single plant of comparable size and developmental stage per pot, facilitating the uniformity of experimental plants. 

### 4.4. Saline Irrigation Treatments

At sixth-leaf stage, plants were exposed to two saline irrigations treatments, i.e., 0 mM NaCl (tap water) and 400 mM NaCl. NaCl concentration was gradually developed in pots with an increment of 80 mM until 400 mM NaCl. Thereafter, concentration of 400 mM NaCl was kept constant throughout the experimental period. After every-fourth irrigation, a nutrient solution was added to irrigation water at a concentration of 1:200 (Hornum, Næring, Roskilde, Denmark) in each pot. Pots were irrigated to 90% of pot water holding capacity to avoid drought stress in plants. 

### 4.5. Growth and Yield Measurements

After eighty-five days of germination, plants were harvested and shoot length and panicle length were measured. Shoot were oven dry for 48 h at 70 °C and shoot biomass was recorded. Grains were collected from panicle to record 100-grains weight and grain yield/plant.

### 4.6. Physiological Measurements

Leaf photosynthetic rate (A_n_) and stomatal conductance (g_s_) were measured from the upper canopy of fully mature leaves (two leaves from one plant in each experimental unit) using a portable photosynthetic system (CIRAS-II, PP System, UK). The measurements were performed from 10:00 to 13:00 h at a CO_2_ concentration of 400 μmol mL^−1^, chamber temperature 28.5 °C and photon flux density of 1200 μmol m^−2^ s^−1^ [20]. 

### 4.7. Measurements of Water Relations

To determine leaf water potential (Ψ_leaf_), the same leaves used for physiological measurement were detached from plant and wrapped in polythene bag. Leaves were immediately put in pressure chamber (Model 3000F01H12G2P40, Soil Moisture Equipment) to measure leaf water potential (Ψ_leaf_). For osmotic potential, the same leaves (used for measuring Ψ_leaf_) were wrapped in aluminum foil and then dipped immediately in liquid nitrogen. Leaves were stored at −80 °C until analysis of osmotic potential (Ψ_π_). Before measuring Ψ_π_, the leaves were first allowed to thaw for 15 min. and then squeezed with forceps to extract sap. Thereafter, osmotic potential was determined from the extracted sap using psychrometer (C-52 sample chambers, Wescor Inc., Logan, UT, USA), which was connected to a datalogger (Wescor’s Dew Point Microvoltmeter, model HR-33T). 

Leaf turgor potential (Ψ_p_) was calculated by using Equation (1), assuming leaf matric potential to be zero.
(1)$$\Psi_{p}=\Psi_{leaf}-\Psi_{\pi }$$

Leaf relative water content was measured as Equation (2):(2)$$\mathrm{Relative}\;\mathrm{water}\;\mathrm{content}\;(\mathrm{RWC})=\frac{\left(\mathrm{FW}-\mathrm{DW}\right)}{\left(\mathrm{TW}-\mathrm{DW}\right)}\times 100$$

Harvested leaf was cleaned with tissue paper to measure fresh weight (FW). Thereafter, leaves were immerged into distilled water for 2 h to get turgid weight (TW) and then dried in an oven at 85 °C for 24 h to obtain dry weight (DW) [20].

### 4.8. Leaf ABA Determination

For the determination of leaf ABA, 30 mg of leaf sample was crushed in liquid nitrogen using mortar and pestle containing 1 mL of milli-Q water. The suspension was added to Eppendorf tubes, homogenized using end-over-end rotatory shaker for 24 h at 4 °C and then centrifuged at 10,000× *g* at 4 °C. Clear supernatant was collected in another Eppendorf tube and stored at 4 °C until analysis. Leaf ABA concentration was then determined through an enzyme-linked immunosorbent assay (ELISA) using a monoclonal antibody for ABA (AFRC MAC252) [58]. 

### 4.9. Enzyme Activity Signature of Antioxidant Metabolism

For determination of central antioxidative enzyme metabolism, protein was extracted as described in Jammer et al. [49]. In brief, leaf samples were ground with pestle and mortar in liquid nitrogen. The ground material was collected and approximately 500 mg of plant material was weighed accurately in 2 mL of 12 Eppendorf tubes. 1 mL of extraction buffer (40 mM TRIS-HCl pH 7.6, 3 mM MgCl_2_, 1 mM Ethylenediaminetetraacetic acid (EDTA), 0.1 mM phenylmethane sulfonyl fluoride (PMSF), 1 mM benzamidine, 14 mM β-mercaptoethanol, 24 μM nicotinamide adenine dinucleotide phosphate (NADP)) was added in each tube. The total volume of extract was dialyzed as described in Jammer et al. [59].

Aliquots of dialyzed plant extracts were shock frozen in liquid nitrogen and kept at −20 °C for later analysis.

Activity signature of key antioxidative enzymes such as superoxide dismutase (SOD; EC:1.15.1.1), catalase (CAT; EC:1.11.1.6), peroxidase (POX; EC:1.11.1.11), glutathione reductase (GR; EC:1.8.1.7), and dehydroascorbate reductase (DHAR; EC1:1.8.5.1) were determined according to Fimognari et al. [60].

Enzyme activities were measured in a semi-high throughput manner using 96-well ultraviolet (UV) transmissive flat-bottomed microtiter plates and the BioTek plate reader Synergy 2. The total reaction volume for all the assays was 160 μL per well. The de- or increase of substrate or product compounds (respectively) was monitored by the change in absorbance at a specific wavelength and the linear phase of compound conversion was used to calculate the enzyme activity in nkat g fresh weight^−1^. For data evaluation the Biotek software Gen5 was used. All assays were carries out in triplicates, for control reactions substrate was omitted.

### 4.10. Leaf Na^+^ and K^+^ Contents

Leaves were detached on 55 days after germination, rinsed quickly with milli-Q water, blotted dry with tissue paper, wrapped in aluminum foil and dipped in liquid nitrogen. Thereafter, leaves were stored at −80 °C until analysis. Leaves were added in Eppendorf tubes. Sap was extracted by crushing leaves using a stainless-steel rod with tapered end. The sap was collected in Eppendorf tubes by Gilson pipette and centrifuged at 10,000× *g* for 5 min [60]. The leaf sap was diluted as required by adding Milli-Q water to determine Na^+^, K^+^ concentration by ion exchange chromatography using a Metrosep C4-100 analytical column (Metrohm AG, Herisau, Switzerland) (4 × 125 mm, 1.7 mM nitric acid or 0.7 mM dipicolinic acid eluent).

### 4.11. Colonization of PsJN in Rhizosphere and Root Interior

Colonization of PsJN from rhizosphere soil and root interior was measured using protocols described in our previous studies [47]. Rhizosphere PsJN colonization was determined by collecting rhizosphere soil from quinoa roots. Slurry was developed by mixing 5 g of rhizosphere soil with 15 mL of 0.9% (*w/v*) NaCl solution. After sedimentation of soil particles, serial dilutions up to 10^−6^ were plated onto selective LB medium containing spectinomycin (100 mg mL^−1^), 5-bromo-4-chloro-3-indolyl- β-D-glucuronide (100 mg mL^−1^), and isopropyl-β-D-galactopyranoside (100 mg mL^−1^). The plates were incubated at 28 °C for 4–5 days and blue colonies were counted to determine the colonisation value.

For root and shoot colonization, tissues samples were surface sterilized (as described above for seed surface disinfection). Thereafter, root and root samples were crushed using sterilized mortar and pestle containing 12 mL of 0.9% NaCl solution. The suspension was homogenized by mixing using end over end rotatory shaker for 30 min. After settling the tissues particles, serial dilutions were spread on selective LB medium. The plates were incubated at 28 °C for 4–5 days and blue colonies were counted to determine the colonization value [61].

### 4.12. Osmoadaptation Assay

The salt-tolerance ability of the bacterial strain (*Burkholderia phytofirmans* PsJN) was determined by conducting osmoadaptation assay at seven salinity levels, i.e., 0, 50, 100, 200, 400, 800 and 1000 mM NaCl solutions in LB media. Firstly, 15 mL of each NaCl solutions (in triplicate) were taken in 30 mL test tubes. Then, the tubes were inoculated with 10 µL of freshly prepared inocula after sterilizing. In addition, 10 µL of broth were added in tubes as un-inoculated control. The tubes were incubated in an orbital shaking incubator at 28 ± 1 °C and 180 rpm. After 48 h of incubation, bacterial growth was assayed by measuring optical density using spectrophotometer (GENESYS 10, Thermo Scientific, Waltham, MA, USA) at λ 600 nm.

### 4.13. Statistical Analysis

The experiment was conducted in complete randomized design with four replications of each treatment. The data was subjected to analysis by two-way analysis of variance (ANOVA) using R version 3.6.3 (R Development Core Team, 2016). The data was presented as mean of four replicates ± S.E., significance between treatments was checked at *p* ≤ 0.05. Regression of some variables were also drawn using Microsoft excel 2011. PCA was also performed using R, all treatments and variables were included in the analysis.

## 5. Conclusions

In conclusion, saline irrigation provoked a decrease in plant growth compared with non-saline irrigation; however, PsJN inoculation activated mechanisms of osmotic adjustment and antioxidant defense to maintain comparatively higher photosynthetic rate, ionic homeostasis and moderate water relations. This was confirmed by PCA analysis where control and PsJN treatments clustered in opposite side of PC2. These findings emphasize that PGPB inoculation could alleviate salinity stress and be potential tools for sustainable agriculture in the future. 

## Figures and Tables

**Figure 1 plants-09-00672-f001:**
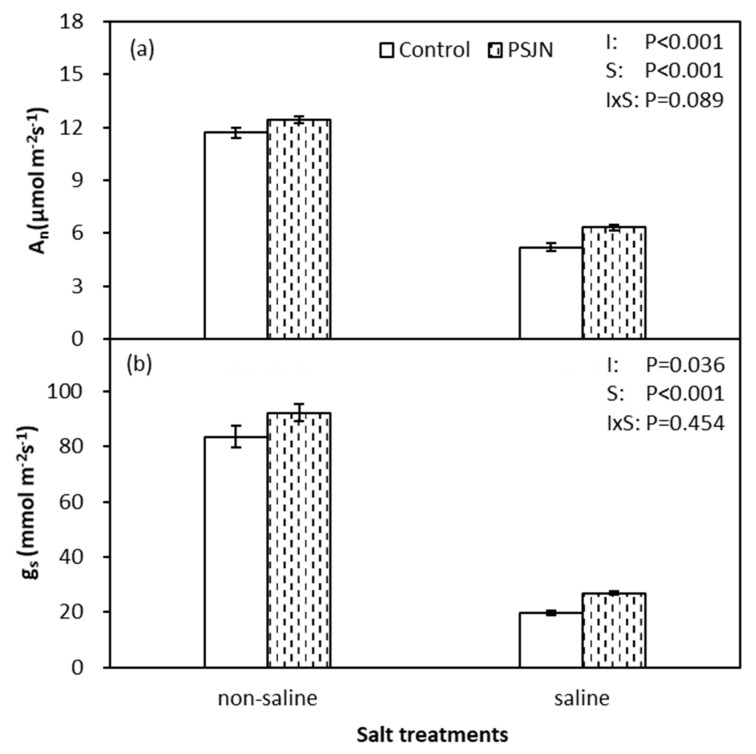
Photosynthetic rate (A_n_) (**a**) and stomatal conductance (g_s_) (**b**) of quinoa leaves affected by bacterial inoculation in non-saline (0 mM NaCl) and saline (400 mM NaCl) irrigations. PsJN indicates *Burkholderia phytofirmans* PsJN. I and S indicate inoculation and salinity treatments, respectively, and I × S indicates the interaction. Error bars indicate standard error (S.E., *n* = 4).

**Figure 2 plants-09-00672-f002:**
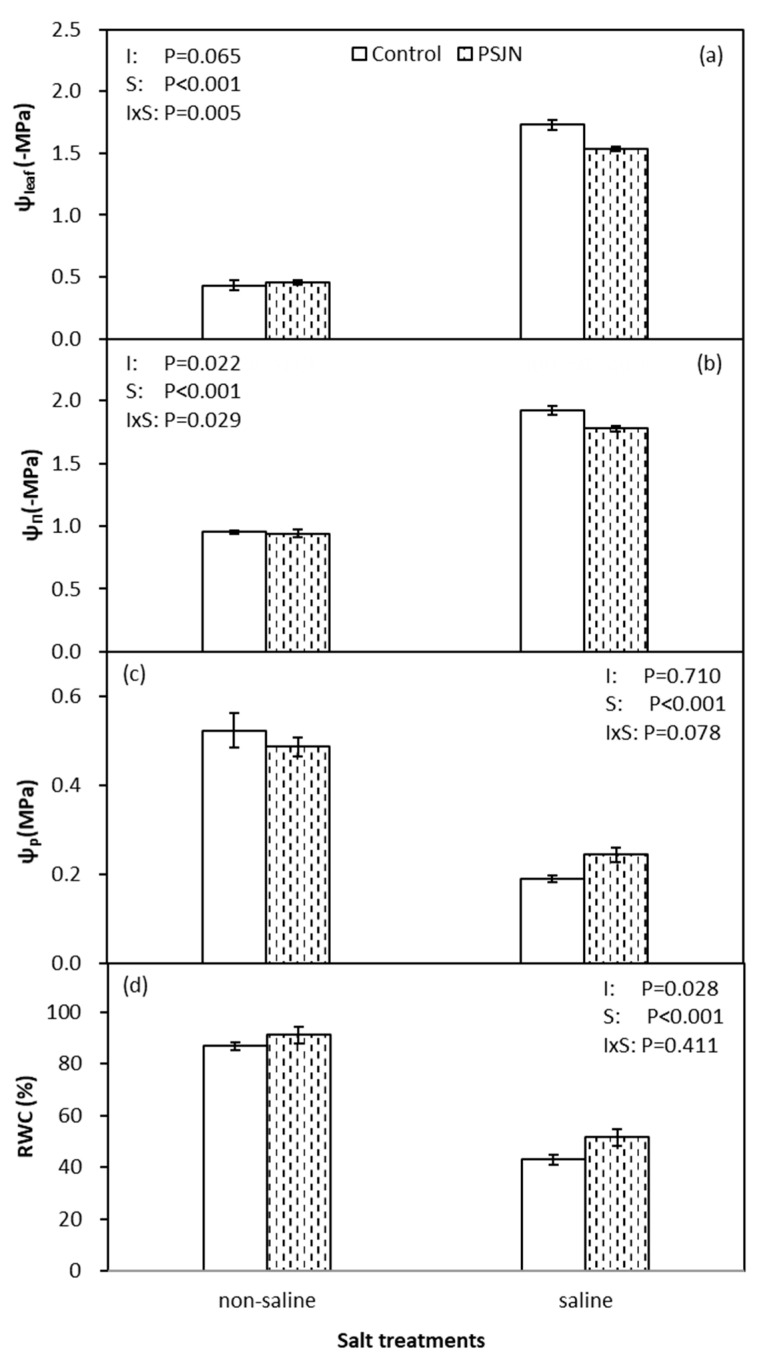
Leaf potential (Ψ_l_) (**a**), osmotic potential (Ψ_π_) (**b**), turgor potential (Ψ_p_) (**c**) and relative water content (RWC) (**d**) of quinoa affected by bacterial inoculation in non-saline (0 mM NaCl) and saline (400 mM NaCl) irrigations. PsJN indicates *Burkholderia phytofirmans* PsJN. I and S indicate inoculation and salinity treatments, respectively, and I × S indicates the interaction. Error bars indicate S.E. (*n* = 4).

**Figure 3 plants-09-00672-f003:**
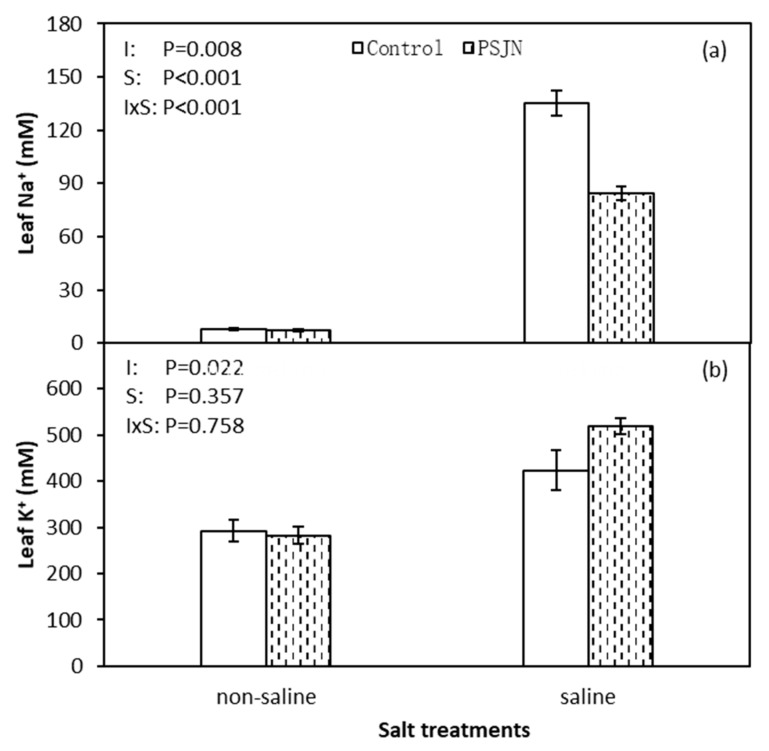
Leaf Na^+^ (**a**) and K^+^ (**b**) concentration of quinoa leaves affected by bacterial inoculation in non-saline (0 mM NaCl) and saline (400 mM NaCl) irrigations. PsJN indicates *Burkholderia phytofirmans* PsJN. I and S indicate inoculation and salinity treatments, respectively, and I × S indicates the interaction. Error bars indicate S.E. (*n* = 4).

**Figure 4 plants-09-00672-f004:**
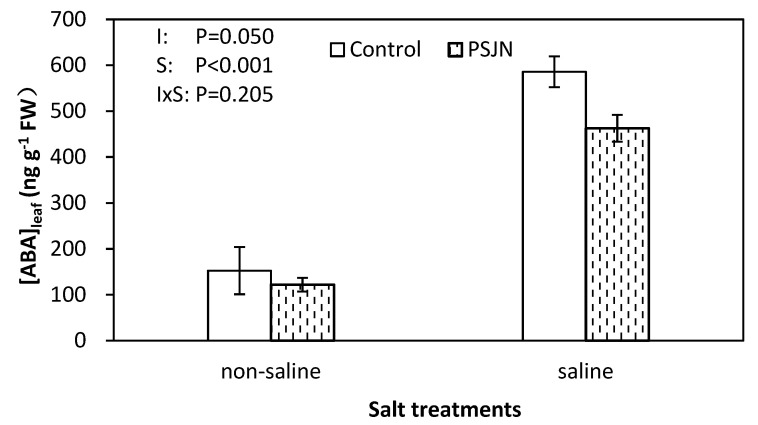
Leaf abscisic acid (ABA) concentrations of quinoa leaves affected by bacterial inoculation in non-saline (0 mM NaCl) and saline (400 mM NaCl) irrigations. PsJN indicates *Burkholderia phytofirmans* PsJN. I and S indicate inoculation and salinity treatments, respectively, and I × S indicates the interaction. Error bars indicate S.E. (*n* = 4).

**Figure 5 plants-09-00672-f005:**
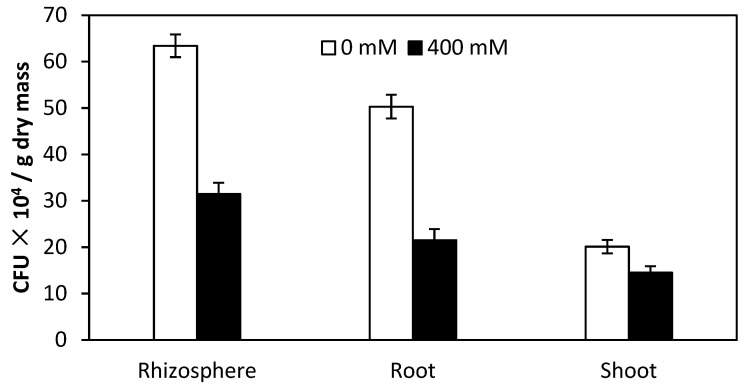
Colony-forming units (CFU) of *Burkholderia phytofirmans* PsJN in the rhizosphere, root and shoot interior of quinoa irrigated with 0 mM and 400 mM NaCl solution.

**Figure 6 plants-09-00672-f006:**
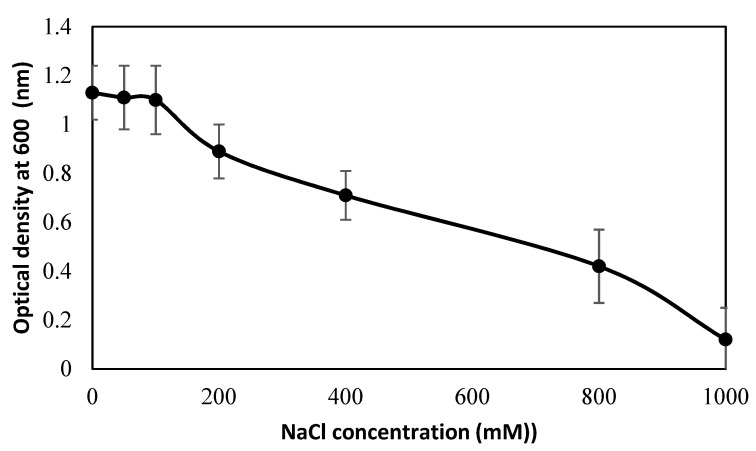
Growth responses of bacterial-strain PsJN observed by measuring optical density (OD) at λ 600 nm under varying salinity levels (ranging from 0 to 1000 mM NaCl solutions).

**Figure 7 plants-09-00672-f007:**
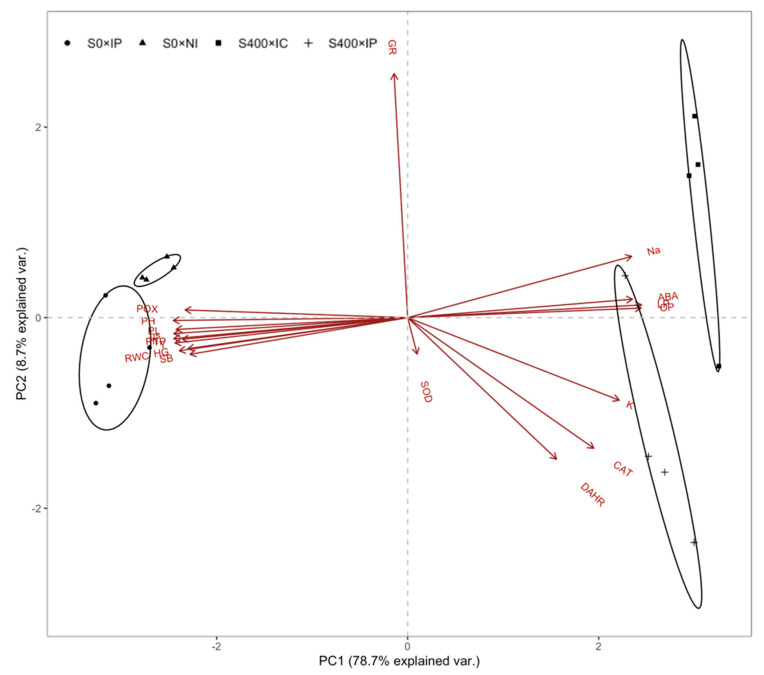
Biplot from PC 1 and PC 2 of principal components analysis (PCA) explained 83.60% of the total variability in data. The angle between two variables indicates correlation between them, acute and obtuse angles indicate positive and negative correlation respectively. For treatments: S0, non-saline (0 mM NaCl); S400, saline (400 mM NaCl); NI, un-inoculation; IP, PsJN inoculation. For variables, PH, plant height; SB, shoot biomass; PL, panicle length; HG, 100-grain yield; Y, grain yield; A_n_, photosynthetic rate; g_s_, stomatal conductance; LP, leaf potential; OP, osmotic potential; TP, turgor potential; Na, leaf Na^+^ concentration; K, leaf K^+^ concentration; ABA, leaf ABA concentration; SOD, superoxide dismutase; CAT, catalase; POX, peroxidase; GR, glutathione reductase; DHAR, dehydroascorbate reductase.

**Table 1 plants-09-00672-t001:** Growth and yield parameters of quinoa influenced by *Burkholderia phytofirmans* PsJN inoculation under different salinity levels (0 mM and 400 mM NaCl).

Attributes	0 mM NaCl		400 mM NaCl		*P* Values		
	Control	PsJN	Control	PsJN	I	S	I × S
Plant height (cm)	140.63 ± 2.07 a	140.13 ± 1.43 a	65.75 ± 1.01 b	73.38 ± 1.91 c	0.052	<0.001	0.031
Shoot biomass (g)	28.70 ± 1.14 a	33.73 ± 1.36 b	19.83 ± 0.59 c	22.19 ± 0.69 d	0.003	<0.001	0.207
Panicle length (cm)	25.75 ± 1.11 a	25.5 ± 1.94 a	8.20 ± 0.21 b	11.13 ± 0.35 c	0.073	<0.001	0.038
100-grain weight (g)	0.32 ± 0.02 a	0.34 ± 0.02 a	0.16 ± 0.01 b	0.19 ± 0.01 c	0.144	<0.001	0.515
Grain yield /plant (g)	13.33 ± 0.26 a	14.43 ± 0.69 b	4.28 ± 0.04 c	6.05 ± 0.10 d	0.002	<0.001	0.386

I and S indicate inoculation and salinity treatments respectively, and I × S indicates the interaction (±standard error, S.E.).

**Table 2 plants-09-00672-t002:** Activity profile of five antioxidant enzymatic signature in quinoa leaf inoculated with *Burkholderia phytofirmans* PsJN and un-inoculated control grown under 0 mM and 400 mM salinity stressed greenhouse conditions represented as a heatmap based on a gradient red-white-blue color scale.

	0 mM		400 mM	
	Control	PsJN	Control	PsJN
DHAR	0.13 ± 0.01	0.12 ± 0.01	0.15 ± 0.01	0.17 ± 0.01
CAT	0.02 ± 0.00	0.03 ± 0.00	0.01 ± 0.00	0.02 ± 0.00
GR	9.27 ± 0.47	7.99 ± 0.47	11.39 ± 1.83	7.17 ± 1.90
POX	0.06 ± 0.00	0.03 ± 0.00	0.1 ± 0.00	0.02 ± 0.00
SOD	10.53 ± 0.91	10.26 ± 2.36	9.86 ± 2.18	10.04 ± 2.28
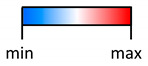				

Dark red color indicates highest, dark blue color lowest values of individual parameters. Numbers are mean ± S.E. (*n* = 4) [nkat/g FW], derived from three biological replicates of three technical replicates each.

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
