# Peer review of "Burkholderia Phytofirmans PsJN Stimulate Growth and Yield of Quinoa under Salinity Stress"

_plants, 2020, doi:10.3390/plants9060672_

Round 1
Reviewer 1 Report
Minor revision
The study is interesting and relevant but Minor revision is required.
It is necessary to revise the paper according to the comments that you can see in the attached file.
1) evidence must be provided that the bacterium is endophytic and may colonize the internal tissues of quinoa plants, the data about СFU in roots detected using select agar seems not sufficient.
2) if it is possible, I recommend including the photos of quinoa plants (inoculated and uninoculated with PsJN) growing both under saline and non-saline conditions.
3) references must be edited according to the requirements of the journal.
4) the text should be check for stylistic and spelling errors.
After revision, the paper may be considered for publishing.

Author Response
We are again thankful to the editor and reviewer 1 for their valuable suggestions and comments for improving the quality of the manuscript. The author’s responses to the comments are attached.

Reviewer 2 Report
Dear authors
The present manuscript sheds light on using plant growth promoters as a significant tool to cope the salinity stress and may be interesting for readers; although i have some concerns and i feel that manuscript have lot of opportunity to be improved before publication on scientific platform. My comments are as below-
- There is a need to revise the English language, for example the first sentence of abstract “One of the greatest challenges facing humanity is to secure sufficient and healthy food for the increasing world population” does not make sense. It can be better as “One of the major challenges is to ensure the sufficient and healthy food availability for the increasing world population in near future”.
- Line 41- Changing climatic conditions and misuse of agricultural land over the last few decades has led 42 to increase in the area of salt affected soils- should include a reference.
- Line 46-60- suits better for discussion section rather than introduction.
- Green manuring is also an effective strategy to improve the soil by managing the pH and may be introduced in introduction a little bit.
- The data should be statistically treated for significant difference in comparison to control and should be mentioned in table and graphs to facilitate the understanding for readers in easier way.
- Line 293 in discussion says “Cultivation of halophytes such as quinoa under salt-affected conditions could be a wise choice to improve global food security”. Although cultivation requires sunny warm climatic conditions and that’s also a major limitation of cultivation of this crop despite of salinity.
- Several osmolytes also accumulate in higher amount during salinity stress and there is no information in present manuscript regarding this issue. They are very important to protect the subcellular functions and cope the stress conditions.
Good luck and best regards
Thank you
Author Response
We are again thankful to the editor and reviewer 2 for their valuable suggestions and comments for improving the quality of the manuscript. The author’s responses to the comments are attached.

Reviewer 3 Report
Authors report a research topic that actually is of great importance. Salinity stress avoid crop growth and yield in many different parts of the world and also in Mediterranean areas. The use of rhizobacteria to improve salt tolerance and growth of quinoa can represent a good start point for improving plant resistance to various abiotic stress. Data should be better discussed, also explaining why authors choose the rhizobacteria and not biostimulant. I recommend a good english revision.
Best regards
Author Response
We are again thankful to the editor and reviewer 3 for their valuable suggestions and comments for improving the quality of the manuscript. The author’s responses to the comments are attached.

Round 2
Reviewer 2 Report
Dear Authors
Thank you for answering the queries and the manuscript have been improved significantly in my opinion. I hereby recommend the manuscript for further processing and publication.
With regards